# Adaptation of the Marine Bacterium *Shewanella baltica* to Low Temperature Stress

**DOI:** 10.3390/ijms21124338

**Published:** 2020-06-18

**Authors:** Anna Kloska, Grzegorz M. Cech, Marta Sadowska, Klaudyna Krause, Agnieszka Szalewska-Pałasz, Paweł Olszewski

**Affiliations:** 1Department of Medical Biology and Genetics, Faculty of Biology, University of Gdańsk, Wita Stwosza 59, 80-308 Gdańsk, Poland; 2Department of Bacterial Molecular Genetics, Faculty of Biology, University of Gdańsk, Wita Stwosza 59, 80-308 Gdańsk, Poland; grzegorz.cech@ug.edu.pl (G.M.C.); catallonia@gmail.com (M.S.); klaudyna.krause@phdstud.ug.edu.pl (K.K.); agnieszka.szalewska-palasz@ug.edu.pl (A.S.-P.); 33P Medicine Laboratory, International Research Agenda, Medical University of Gdańsk, Dębinki 7, 80-211 Gdańsk, Poland

**Keywords:** *Shewanella baltica*, marine bacteria, prokaryotes, cold stress, transcriptomics, gene expression

## Abstract

Marine bacteria display significant versatility in adaptation to variations in the environment and stress conditions, including temperature shifts. *Shewanella baltica* plays a major role in denitrification and bioremediation in the marine environment, but is also identified to be responsible for spoilage of ice-stored seafood. We aimed to characterize transcriptional response of *S. baltica* to cold stress in order to achieve a better insight into mechanisms governing its adaptation. We exposed bacterial cells to 8 °C for 90 and 180 min, and assessed changes in the bacterial transcriptome with RNA sequencing validated with the RT-qPCR method. We found that *S. baltica* general response to cold stress is associated with massive downregulation of gene expression, which covered about 70% of differentially expressed genes. Enrichment analysis revealed upregulation of only few pathways, including aminoacyl-tRNA biosynthesis, sulfur metabolism and the flagellar assembly process. Downregulation was observed for fatty acid degradation, amino acid metabolism and a bacterial secretion system. We found that the entire type II secretion system was transcriptionally shut down at low temperatures. We also observed transcriptional reprogramming through the induction of RpoE and repression of RpoD sigma factors to mediate the cold stress response. Our study revealed how diverse and complex the cold stress response in *S. baltica* is.

## 1. Introduction

Bacteria had evolved multiple mechanisms for efficient adaptation to changes in their habitats. In particular, in addition to limitations in the nutrient supply, environmental bacteria encounter sudden alterations in the basic physical and chemical parameters of their habitats. The aquatic environment constitutes over 70% of the Earth’s surface and hosts a variety of microorganisms providing biomass and energy production. Marine bacteria display a significant versatility in adaptation to variations in environment and stress conditions, including temperature shifts due to seasonal changes and, even more abrupt, water currents transferring bacteria to new locations (vertically and horizontally) [1]. Among other physical stresses, adjustments to low temperatures have been studied in detail using model microorganisms, such as the well-known mesophilic *Escherichia coli* [2] or *Bacillus subtilis* [3], and environmental species such as *Vibrio harveyi* [4,5,6,7]. Adaptation to the abrupt temperature downshift usually involves an acclimatization stage where the gene expression pattern is reprogrammed to shut down unnecessary processes and induce those beneficial for survival, which is followed by resuming of growth at a rate adapted to the altered condition. For mesophilic bacteria, such adaptation typically involves upregulation of the synthesis of cold shock proteins (Csp) which serve as RNA chaperones, transcription modulators and DNA topology organizers [8,9]. As the low temperature affects bacterial membrane stability and permeability, one of the described responses is related to decreased lipid saturation to prevent membrane stiffness [10,11]. Metabolic changes under cold stress involve repression of typical glycolysis and the tricarboxylic acid (TCA) cycle, in favor of secondary and alternate pathways [10]. Accumulation of specific compatible solutes (glycine, betaine, glycerol, trehalose, sucrose, mannitol, sorbitol) plays an important role in cryoprotection, preventing protein aggregation, free radical accumulation and membrane stabilization [12]. In this process, function of the transmembrane transporters is adjusted accordingly.

Marine bacteria inhabiting the temperate and cold climate zones tend to develop psychrotolerant phenotypes, enabling them not only to survive but also thrive at the suboptimal temperatures. This is achieved by traits allowing for optimization of growth, enzyme functioning, motility and macromolecule synthesis [13,14,15]. Cold adaptation that leads to their optimal activity involves changes in the cell membrane structure and fluidity, protein folding with the help of chaperones, and changes in the nucleic acids topology that affect both, replication and gene expression [16,17,18].

The *Shewanella* genus is one of the most abundant γ-proteobacteria in the marine and fresh water environment. Its metabolic versatility and ability to utilize a variety of extracellular electron acceptors is a key feature in its role in the turnover of organic matter, denitrification and bioremediation [19,20]. In addition, *Shewanella* species are known for their spoilage and decomposing potential even at low temperatures, which indicates that their metabolic processes are active under those conditions [21]. Despite of the reported psychrotrophic features of the *Shewanella* genus representatives, the detailed molecular processes underlying their survival and metabolic activity at low temperature has been studied mostly only for the best known species, such as *Shewanella oneidensis*, for the deep-sea dwelling *Shewanella piezotolerans,* and for the Antarctic species *Shewanella livingstonensis* [22,23,24,25], while other *Shewanella* species provided mostly only some information on the cold-adapted enzymes and relevant processes [26,27]. *Shewanella baltica* is a representative of the *Shewanella* genus inhabiting the Baltic Sea, where it is considered as the main player in reducing nitrates at the redox transition interface and its dominant role in denitrification of this habitat was reported [28,29]. A study of the response of *S. baltica* to various stressors revealed that upon nutrient limitation, heat shock and DNA damage these bacteria induce the stringent response—a global reaction to stress—as one of the ways of dealing with the challenging conditions [30].

As the cold stress response strategy may not be consistent in various bacteria and quite often differs between species, discovering how marine bacteria can cope with temperature shifts would provide valuable information on bacterial adaptability. To this end, the question of *S. baltica* response to low temperatures has not yet been resolved. Therefore, in this work we aimed to characterize transcriptional response of *S. baltica* to cold stress in order to achieve a better insight into mechanisms ensuring its adaptation. We found that the general response of *S. baltica* involves downregulation of gene expression after 90 and 180 min after cold stress onset (approximately 70% of differentially expressed genes); however, expression of genes from several groups and pathways was significantly activated. Here, we discuss a possible role of the observed changes in the gene expression profile in adaptation and survival of cold stress exposed *S. baltica*.

## 2. Results and Discussion

### 2.1. Global Changes in the Transcriptome of S. baltica in Response to Cold Stress

To characterize the transcriptional response of *S. baltica* to cold stress we exposed bacterial cells to 8 °C for 90 and 180 min, followed by RNA sequencing (RNA-seq). For selection of genes whose expression has significantly changed, we followed the rules of good practice in gene expression analysis [31]. The main role was to filter genes with distinct changes, therefore, our main data set was composed of genes which had at least a two-fold change in expression (|log_2_FC| > 1) and adjusted *p* value < 0.01 (Appendix A). Due to the limited number of replicates, we decided to use these stringent filtering criteria as a minimum threshold for scoring differentially expressed genes in the first instance.

The most prominent effect of cold stress was downregulation of gene expression (Figure 1A). With the above criteria, we identified 521 genes downregulated after the first 90 min of cold stress, out of which 491 genes were still downregulated after additional 90 min exposure to low temperature, indicating persistent downregulation (Figure 1B). In addition, 92 genes were downregulated exclusively after 180 min of cold stress. In contrast, only 216 genes were upregulated after 90 min, out of which 5 remained upregulated after subsequent 90 min of cold stress exposure (Figure 1C). Sixteen genes were upregulated exclusively only after 180 min of growth at 8 °C.

In summary, the major effect of cold stress in *S. baltica* is a massive decrease in gene expression, which was observed for approximately 70% of differentially expressed gene sets. The same direction in transcriptomic changes that was observed in our study, was reported for *S. oneidensis* [23] and *E. coli* [11] as well. Thus, downregulation of gene expression appears to be a general cold stress response in bacteria.

Since we performed the RNA sequencing experiment in duplicate, it raises a question about robustness and significance of our findings. As stated above, we used stringent criteria for finding genes with significant changes in expression. However, we are aware that despite high coverage (average 783×, Appendix A), the number of replicates allows for correct detection of differentially expressed genes with 40% probability (estimated with RNASeqPower package version 1.28.0 [32] for 783× coverage, 2 replicates and α = 0.05). To verify our RNA-seq data, we tested the expression of 11 selected genes with quantitative reverse transcription PCR (RT-qPCR) assay (Figure 2). The comparison of gene expression patterns obtained from RNA sequencing and RT-qPCR assay revealed that all tested genes show the same direction of change (Figure 2C). This result is higher than the calculated 40% which is probably due to the stringent adjusted *p* value threshold used for filtering of our RNA-seq results. Nevertheless, our RT-qPCR analysis corroborates findings from the RNA-seq analysis. 

### 2.2. Changes in the Metabolic Pathways of S. baltica in the Adaptation and Survival under Cold Stress Conditions

For analysis of metabolic pathways which could be enriched upon cold stress, the KEGGprofile package version 1.30.0 was employed [33]. The three-letter identifier for *S. baltica* OS185 used in our analysis was “sbm”. Kyoto Encyclopedia of Genes and Genomes (KEGG) pathway enrichment in genes up- and downregulated upon cold stress was considered significant for *p* value < 0.01. For numbers of enriched genes, gene names and statistics obtained in the pathway enrichment analysis see the Appendix A. Fold change values of differentially expressed genes are available in Appendix A.

Enrichment analysis showed that only few pathways were upregulated in the cold stress exposed *S. baltica* (Appendix A). After 90 min, there was a significant upregulation of a few metabolic pathways, including glycerolipid metabolism (KEGG: 00561), riboflavin metabolism (KEGG: 00740), pentose phosphate pathway (KEGG: 00030), and aminoacyl-tRNA biosynthesis (KEGG: 00970). Interestingly, flagellar assembly (KEGG: 02040) was also detected as a significantly upregulated cellular process. Additionally, after 180 min the sulfur metabolism pathway became upregulated (KEGG: 00920). However, much more pathways were downregulated upon cold stress (Appendix A). After 90 min, there was a significant downregulation of glycolysis/gluconeogenesis (KEGG: 00010), fatty acid degradation (KEGG: 00071), pyruvate metabolism (KEGG: 00620), propanoate metabolism (KEGG: 00640), C5-branched dibasic acid metabolism (KEGG: 00660), and pantothenate and CoA biosynthesis (KEGG: 00770). Significantly downregulated pathways were assigned to amino acid metabolism, including pathways of valine, leucine and isoleucine degradation and biosynthesis (KEGG: 00280 and 00290), histidine metabolism (KEGG: 00340), and phenylalanine, tyrosine and tryptophan biosynthesis (KEGG: 00400). Downregulation was also observed for a bacterial secretion system (KEGG: 03070). After 180 min, additional pathways became downregulated, including pathways of amino acid metabolism—arginine and proline metabolism (KEGG: 00330) and beta-alanine metabolism (KEGG: 00410), as well as of butanoate metabolism (KEGG: 00650), and nicotinate and nicotinamide metabolism (KEGG: 00760).

Many pathways that were affected by gene expression changes in *S. baltica* in our study, were also reported as affected by cold stress in *S. oneidensis* [23]. For example, downregulation of genes in *S. oneidensis* was dominant among amino acids biosynthesis or energy metabolism pathways. Genes whose products are involved in the fatty acid and phospholipid metabolism or transport/binding proteins were affected as well. The same pathways or similar processes were affected in our study of *S. baltica*. To conclude, *S. baltica* shows a wide range of transcriptional changes in response to the cold stress challenge. Comparison of these changes with changes in other *Shewanella* species may reveal a more general picture of the transcriptomic cold stress response of these bacteria. In subsequent sections, we will discuss a possible role of selected gene expression changes in potential adaptation and survival strategies of *S. baltica* undergoing cold stress.

### 2.3. Transcriptional Reprogramming to Adjust to Cold Stress

Transcriptional regulation in bacteria is typically controlled at the transcription level, which would allow the quickest response with a minimum energy loss. Thus, bacteria have developed numerous strategies to regulate transcription initiation in response to various challenges. One of the most efficient ones is using various sigma subunits of the RNA polymerase to alter the transcription machinery’s capacity and redirect its functioning towards specific promoters, operons and regulons. Some sigma factors are directed to counteract and protect bacteria from the environmental challenges, such as heat stress, nutrient limitations and others, and their number usually correlates with the variability of habitats and metabolic flexibility [34]. Transcriptional reprogramming through various sigma usage would be a sign of bacteria dealing with stress conditions. In fact, we observed that the gene encoding the housekeeping sigma 70, *rpoD* (Shew185_3128), was downregulated upon cold stress, as was the case for genes coding for sigma 32 (Shew185_4161), sigma 54 (Shew185_0686) and sigma E (Shew185_1239—*rpoE*). A significant upregulation was observed for the Shew185_2636 gene coding for the extracytoplasmic function (ECF) subfamily RNA polymerase sigma 24 factor (*rpoE2*) and for Shew185_2919—the flagellar biosynthesis sigma factor (Appendix A). Changes in gene expression revealed by the transcriptome analysis were confirmed for genes chosen for further qPCR verification: *rpoE*, *rpoE2* and *rpoD* (Figure 2). Notably, for the general transcription response, expression of the gene responsible for synthesis of the stringent response alarmones, ppGpp and pppGpp, had increased (Appendix A). The possible accumulation of (p)ppGpp would result in favoring of alternative sigma factors’ association with the RNA polymerase core for upregulation of stress-induced gene expression; in case of the studied cold stress—this would mainly concern the ECF factors and flagellar sigma factors.

ECF sigma factors are responsible for integrating signals from the outer membrane and its components, such as proteins (transporters, porins) and lipopolysaccharides, as well as changes in the concentrations of compounds outside the cells [35,36,37]. Their response to the envelope stress is often regulated by the numerous anti-sigma factors which, being anchored as transmembrane proteins, serve as signal transduction systems, releasing sequestered ECF sigma factors to activate relevant regulons [38,39]. Interestingly, expression of one of the annotated *S. baltica* anti-sigma E factors, RseA (Shew185_1240), was downregulated (Appendix A). Many bacteria with more complex life styles have multiple ECF sigma factors, which together with their regulators constitute a complicated control system [38]. In the *Shewanella* genus, the ECF factors were described so far only for *S. oneidensis*, where sigmas from the ECF-like family were identified [40]. The two *S. baltica* ECF sigma factors that were observed in our analysis to have altered expression upon cold stress are S. *oneidensis rpoE* and *rpoE2* orthologs (Shew185_1239 and Shew185_2636, respectively). The *S. oneidensis rpoE* was described as a sigma factor responsive to high temperatures and osmotic stress, while *rpoE2* was not reported to be required for a temperature stress but was involved in the oxidative stress response [40]. One of the genes transcribed by the RpoE2-RNA polymerase, the periplasmic glutathione peroxidase, is important for resistance to the oxidative stress. In our studies, *S. baltica* Shew185_3037 gene coding for the periplasmic glutathione peroxidase and Shew185_3036 encoding a ferrochelatase, present in the same operon, displayed an increased transcription, corresponding with *rpoE2* upregulation. These data indicate that upon cold stress *S. baltica* responds by inducing mechanisms for protection against oxidative stress. Low temperatures promote increase in the reactive oxygen species due to increased oxygen solubility [10,41]; thus, counteraction against reactive oxygen species (ROS) formation is an important strategy to maintain the metabolic and genomic stability. It was reported that at low temperatures oxidative metabolism is repressed in order to decrease ROS production [10,42], however, in our studies we have not observed such an unequivocal response in carbon metabolism (while transcription of some of the TCA cycle genes was downregulated or remained constant, it did not hold true for other enzymes of this cycle or for the glycolysis pathway genes where some of them were upregulated). Thus, we can conclude that in response to cold stress, *S. baltica* activates oxidative stress protection in the *rpoE2*-dependent pathway. It is worth mentioning that stimulation of the protective system against ROS also serves as a defense mechanism against antibiotics in *S. oneidensis.* This process is linked to cellular H_2_S production [43]. Indeed, it was suggested that H_2_S plays an important role in prokaryotic biology, however, our understanding of this process is still limited [44]. In this study, we observed changes in gene expression of enzymes involved in the sulfur metabolism (including H_2_S production) and these findings are further discussed in Section 2.7.

### 2.4. Expression of S. baltica Cold-Induced Proteins in Response to Low Temperatures

A typical bacterial response to a temperature downshift is associated with an increase in the production of cold induced proteins (Cip, also referred to as Csp—cold shock proteins) [45]. These small proteins have a conserved nucleic acid binding domain and are widespread in nearly all tested bacteria (independently of their habitat) [46]. These proteins mainly play a role of nucleic acid chaperones, affecting transcription initiation under conditions when stabilization of mRNA due to low temperature would hamper gene expression; however, some Csp may be active under other stress conditions as well [46,47]. Two of those proteins have been identified in *S. oneidensis* as responding to cold stress, mostly at the acclimatization stage [23], yet their cellular level may remain elevated for longer periods of time. Orthologs of those proteins in *S. baltica* are encoded by Shew185_1705 and Shew185_1464; both of them were significantly upregulated at 90 and 180 min after onset of cold stress in our study (Appendix A). This indicates that under these conditions *S. baltica* adjusts and regulates the level of gene expression also by affecting the mRNA stability.

### 2.5. Expression of Flagellar Assembly-Related Genes in Cold Stress Exposed S. baltica

Bacterial motility is one of the key features to ensure a proper response to environmental stressors. The most common organelles in motile bacteria are flagella, necessary not only for chemotaxis but also for adherence, biofilm formation, virulence or such processes as reduction of insoluble minerals [48,49,50,51]. A typical flagellum structure is well described and conserved among bacteria, and consists of over 20 proteins forming specific structures: the hook and basal body that anchor the flagellum to the membrane, and the external filament [52,53,54]. The energy that bacteria use to produce flagella is substantial, as it requires, among others, synthesis of thousands of the flagellin molecules to build a filament; moreover, action of the flagellum is based on the proton pumps serving as motor force generators. Such a high energy usage comprises a large burden on bacterial metabolism, and thus the process of flagellar assembly, as well as its rotation, requires a precise regulation [55]. This is achieved by a sequential cascade of gene transcription. In bacteria with a polar flagellum, the first level of transcription control is usually mediated by promoters dependent on sigma 54 and transcription regulators (among others, sigma subunit specific for flagella). Subsequent expression tiers are under control of these factors and encode flagellar structural genes [56]. Bacteria of the *Shewanella* genus, although highly diverse in terms of habitats, are usually motile with one polar flagellum. As described for the group representative, *S. oneidensis*, most components of the flagellum (except for the motor protein genes) are encoded by genes located in one chromosome segment [57]. A similar genomic set-up is present in *S. baltica* (based on the sequenced genomes). We observed that the flagellar structural genes (encoding the basal body, rod, filament, as well as the anchoring complex) were upregulated in *S. baltica* after the onset of cold stress, with higher expression usually at the 90 min time point (Figure 3). Such an increased expression was also noted for the regulatory genes, e.g., the flagellar biosynthesis sigma factor, *fliA*. The temperature dependent induction of flagellar synthesis has been reported for various bacteria, including pathogenic *Listeria monocytogenes* [58], where the Csp proteins regulate the flagellum gene expression [59], and also in *Yersinia enterocolitica* [60]. Flagellar assembly was induced in the extremophile bacterium, *Thermoanaerobacter tengcongensis* [55] as part of a cold stress response; also, in an Antarctic bacterium, *S. livingstonensis,* cold adaptation was accompanied by flagellar genes’ overexpression [24]. Notably, the increased expression of genes encoding flagellar proteins was also observed in *E. coli* after a temperature downshift [61]. These data indicate that the motility of cold exposed flagellated bacteria intensifies, based on the upregulation of transcription of flagellar genes and those involved in the flagellar assembly. Interestingly, this adaptation does not seem to correlate with the bacterial life style or habitat. As observed by our transcriptomics data, *S. baltica* follows this pattern; the RT-qPCR analysis confirmed this observation for the flagellar basal body rod protein-encoding gene (*flgF)* whose expression level was increased at 180 min time point, while the *fliA* transcript level was elevated mostly at 90 min, which corresponds with the role of *fliA*-encoded sigma factor to promote the flagellar gene expression (Figure 2A). A biological sense of the induced flagella production and bacterial motility during temperature decrease has been proposed to be related to the need for an increased chemotaxis towards energy sources, the usage of unusual electron acceptors, and also as a response to higher viscosity of liquids in the cold [58,62,63]. Besides that, flagella promote biofilm formation, which can be also important for survival of *S. baltica* and its ability to grow on the surface of fish and sea-food, causing its spoilage. Moreover, flagella can serve as a secretion system which can be important for optimal survival at low temperatures.

### 2.6. Downregulation of a Type II Secretion System in Response to Cold Stress

All bacterial species transport a great number of proteins across the cytoplasmic membrane, from the cytosol to specific outer locations. Lipoproteins, periplasmic proteins, cell wall-associated proteins, outer membrane or extracellular proteins are all subjected to secretion in growing cells [45]. Processing and translocation of preproteins are pointed out to be reduced in the early phase in bacteria exposed to cold stress conditions, as well as in the cold adapted cells [45].

Secretion system in the Gram-negative bacteria consists of several distinct systems mediating secretion of proteins for example involved in biogenesis of pili and flagella, nutrient acquisition, virulence, adherence and biofilm formation, or efflux of drugs and other toxins [64,65]. One of these systems, the type II secretion system (T2SS), is responsible for translocation of exoproteins through the outer membrane into the extracellular *milieu*. In *S. oneidensis* and *Shewanella putrefaciens*, this system is directly involved in extracellular translocation of outer membrane cytochromes, as well as exoproteins engaged in dissimilatory reduction of insoluble metal oxides, electron transport and anaerobic respiration [66,67]. The multi-component T2S system consists of a cytoplasmic ATPase, several inner membrane proteins, a periplasmic pseudopilus and a secretin pore embedded in the outer membrane [68] (Figure 4). Components of this system belong to a family of general secretory pathway (Gsp) proteins found among all bacterial species. Secretion of exoproteins with T2SS is a two-step process [69]. First, the substrate exoprotein must be translocated across the inner membrane by either SecA and signal recognition particle (Sec-SRP) translocon, or the twin-arginine targeting (Tat) translocon. Second, after reaching the periplasm, the exoprotein is exported by the T2S system across the outer membrane.

Our transcriptomic study revealed that in *S. baltica*, the T2S and Sec-SRP system components are regulated in response to cold stress exposure (Figure 4). We found that 11 genes of the large Gsp operon encoding T2SS components were significantly downregulated after 90 and 180 min of temperature downshift. The observed downregulation affected genes coding for all components of the T2S system, including the GspD secretin (Shew185_0149), all inner membrane proteins (IMP)—GspC, GspF, GspG, GspH, GspI, GspJ, GspK, GspL and GspM (Shew185_0148, Shew185_0151, Shew185_0152, Shew185_0153, Shew185_0154, Shew185_0155, Shew185_0156, Shew185_0157 and Shew185_0158, respectively) and the GspE ATPase (Shew185_0150), and the extent of downregulation was greater over time (Figure 4). Our results indicated that the entire T2SS is affected by low temperature and this may probably lead to a serious reduction in protein secretion mediated by this system. To the best of our knowledge, downregulation of type II secretion system in response to cold stress was not previously reported for the *Shewanella* genus. 

The translocation of unfolded proteins from the cytoplasm to periplasm, where the folding occurs before the protein reaches the T2S system, is mediated by the Sec-SRP translocase system located in the inner cell membrane. One key element of this process is the SecB chaperone responsible for post-transcriptional delivery of preproteins to the Sec translocon [69]; the other one is SRP, composed of the Ffh protein and 4.5S RNA, which mediates co-translational targeting of preproteins [70]. Fully folded proteins in the cytoplasm are transported across the plasma membrane to the periplasm by the Tat system [69] and T2SS is the only known secretion system for the Tat-delivered exoproteins [65]. Upon cold stress exposure, expression of genes encoding components of the general secretory pathway (Sec) and HlyD family secretion proteins were reported to be either repressed or unaffected in *S. oneidensis* [23], but expression of *tatA*, *tatB* and *tatC* genes encoding components of the Tat-dependent protein translocon were shown to be induced by low temperature [23]. In our study, we observed a significant downregulation of expression of the SecB-encoding gene (Shew185_0045) upon cold stress (Figure 4), suggesting that targeting of exoproteins to the Sec translocon for co-translational transport across the inner membrane may be highly temperature-dependent in *S. baltica*. However, we observed a transcriptional downregulation in cold exposed *S. baltica* only for one component of the Sec-SRP translocon itself, the SecD/F (Shew185_2804), while several other of its components were significantly upregulated. We also found upregulation of the *ffh* gene expression (Shew185_1253), i.e., of the gene coding for the signal recognition particle protein. Expression of the Tat system components was also slightly affected by low temperature—the TatC (Shew185_0414) encoding gene was upregulated after 90 and 180 min of cold stress, while expression of TatA (Shew185_0416) and TatB (Shew185_0415) encoding genes was downregulated (Figure 4). In our study, expression of the Tat translocon-encoding genes was only slightly regulated upon cold stress at the transcriptional level, but as the proteins delivered to the periplasm via this route are secreted by the T2S system, secretion of these exoproteins may also be affected by the low temperature stress.

In summary, our study confirms that cold stress conditions affect protein transport across membranes in *S. baltica*. Transcriptional data suggest that transport of the Sec-SRP-dependent proteins may be limited at least to some extent due to downregulation of the SecB chaperone gene expression, required for appropriate targeting of preproteins to the Sec translocon; downregulation of the entire T2S system expression may result in substantial restriction of the exoprotein secretion. As the T2SS transport is reduced at the transcriptional level throughout the entire period of our experiment, such downregulation may be considered as both, an early response and an adaptation of *S. baltica* to low temperature conditions.

### 2.7. Upregulation of the Sulfur Assimilatory Pathway upon Cold Stress

Sulfur is an element essential not only for bacteria, but for life in general. Various oxidation states of sulfur—ranging from (+6) in sulfate SO_4_^2−^, through (+4) in sulfite SO_3_^2−^, to (−2) in sulfide H_2_S, implicate the vast reduction/oxidation potential of this element. In general, sulfur metabolism comprises of sulfur oxidation and sulfur reduction pathways. The sulfur oxidation pathway, also known as the sulfur oxidation pathway (SOX) system, is found in photosynthetic and non-photosynthetic sulfur-oxidizing bacteria [71]. The sulfur reduction pathways comprise of two contradictory metabolic routes leading to either assimilation or dissimilation of sulfur-rich compounds. This attribute accounts for the great potential of bacteria to adapt their metabolism to distinct or changing environmental conditions [72,73,74].

In our study, we found that the transcriptional response of *S. baltica* to cold stress includes significant upregulation of expression of genes assigned to the assimilatory branch of the sulfate reduction pathway (Figure 5). In general, this process is energy consuming and produces reduced sulfur compounds which are further utilized for biosynthesis of sulfur-containing amino acids. After 90 min of cold stress, we observed upregulation of three genes involved in this pathway encoding three subunits of the ABC sulfate transporter—the ATPase (Shew185_1063) and two inner membrane subunits (Shew185_1064 and Shew185_1065). Upregulation of these genes was still present after 180 min of the onset of cold stress. The sulfate ABC transporter is involved in import of sulfates from the environment. In the assimilatory pathway, sulfate is activated with ATP by the CysND sulfate adenylyltransferase (Shew185_0928, Shew185_0929,) and forms adenylyl sulfate (APS), which is next converted into 3′-phosphoadenylyl sulfate (PAPS) by the CysC adenylyl-sulfate kinase (Shew185_0931). PAPS is then reduced to sulfite by the CysH phosphoadenosine phosphosulfate reductase (Shew185_0921). Finally, sulfite (SO_3_^2−^) is reduced to sulfide (H_2_S) by the CysIJ assimilatory sulfite reductase (Shew185_0919) [72,73,74,75,76]. All of the genes encoding these enzymes were significantly upregulated in *S. baltica* after 90 and 180 min of cold stress (Figure 5).

Hydrogen sulfide (H_2_S) is not directly excreted, but is used to synthesize L-cystine (with accompanying acetate release) and/or L-homocysteine (with accompanying succinate release). Genes encoding enzymes catalyzing these reactions, CysK (Shew185_2117) and MetB (Shew185_0536) were also upregulated in cold stressed *S. baltica*.

It is interesting that the only downregulated genes in the sulfur metabolic pathway were those encoding the tetrathionate reductase—*ttrB* (Shew185_3734), and thiosulfate/3-mercaptopyruvate sulfurtransferase (Shew185_1160), i.e., enzymes playing a key role in tetrathionate and thiosulfate metabolism. Under anaerobic conditions there is a switch of sulfur metabolism to the dissimilatory pathway in which APS is directly reduced to sulfite and then to sulfide (H_2_S) by a dissimilatory sulfite reductase (pathway not shown on the scheme of sulfur metabolism in Figure 5). The dissimilatory pathway of sulfate reduction leads to anaerobic respiration in which many sulfur compounds may serve as the terminal electron acceptors of the respiratory chain. It was reported that the extraordinary ability to use a variety of respiratory electron acceptors, including iron (III), manganese (IV), chromium (VI), uranium (VI), nitrate, elemental sulfur, sulfite, thiosulfate, and tetrathionate, is the hallmark of the *Shewanella* genus [75,76,77]. A molecular mechanism of anaerobic respiration involving inorganic sulfur compounds in *S. oneidensis* was suggested, but it is still poorly understood [76].

The dissimilatory sulfur pathway itself was not found in our analysis as affected in any direction. However, *S. baltica* was found to produce large quantities of inorganic sulfide (H_2_S) as a consequence of fish spoilage during prolonged on-ice storage. This phenomenon has a great implication for the fishery industry [21]; the *Shewanella* genus was named after a scientist working in the fisheries microbiology, aptly named J. M. Shewan [78]. It has to be mentioned that H_2_S is released from bacterial cells only through anaerobic respiration of the dissimilatory sulfur pathway. Therefore, there is a link between low temperature conditions and the metabolic switch to the dissimilatory sulfur pathway. It was also reported that anaerobic respiration enzymes, such as the sulfite reductase and sulfate reductase, are significantly elevated at the transcriptional level in *S. algae* 2736 cells after high salt exposure [79].

Thus, it may be possible that *S. baltica* has a good reason to keep the tetrathionate and/or thiosulfate conversions to sulfite downregulated, so these compounds could accumulate and serve as electron acceptors in the anaerobic respiration. The question is how can the cold stress or high salt stress lead to anaerobic respiration through the dissimilatory sulfur pathway? Cold or high saline conditions lead to anaerobic conditions in the complex marine environment. Thus, cold stress might be a signal for *S. baltica* cells that anaerobic conditions are possible to attain. There is a high flow of fresh water from rivers but a limited sea water exchange with the Northern Sea in the Baltic Sea; deep water is ventilated only by large perturbations. Low temperature may cause a vertical gradient of salinity, which suppresses adequate ventilation of deep water, resulting in decreased oxygen and increased hydrogen sulfide concentrations [80].

### 2.8. Modified Expression of Genes Responsible for Membrane Integrity and Composition

Properties of cellular membranes depend on the membrane structure and composition, influencing such important processes as transport of nutrients and ions, solute uptake, and osmotic pressure [81]. Proper membrane fluidity is required to maintain its structure and function. Thus, during cold stress, in addition to the response to the envelope stress by inducing sigma E-transcribed genes, bacteria have to deal with decreased membrane fluidity causing membrane stiffness [82]. Composition of the membrane fatty acids is one of most important features responsible for membrane fluidity. Bacteria respond to lowered temperature by increasing the proportion of unsaturated fatty acids (UFA) in the membrane, e.g., by desaturation of fatty acids in phospholipids or by incorporation of unsaturated instead of saturated fatty acids in the membrane structures [83,84]. On the gene expression level, these adaptations can be achieved by various ways—in *E. coli* and many other Gram-negative bacteria, the *lpxP* gene is induced by stress and its product is responsible for attaching an unsaturated fatty acid (the palmitoleate) instead of a saturated laurate to lipid A [85]. In *Bacillus subtilis*, *Listeria monocytogenes* and some other Gram-positive bacteria the proportion of branched fatty acids increases upon cold stress due to their elevated synthesis [86]; moreover, expression of the fatty acids desaturase gene, *des*, is induced [84,87]. Synthesis of UFA can be achieved through two pathways—anaerobic, dependent on the FabA and FabB enzymes (as shown in *E. coli*), and aerobic—by the Des enzyme [84,88]. Expression of the *fabA* and *fabB* genes was shown to be under control of an activator (FadR) and a repressor (FabR) [89]. Moreover, increased cold adaptation is correlated with the synthesis of polyunsaturated fatty acids (PUFA), with two main genes playing an important role in this process, *pks* and *pfaB* [90]. Synthesis of UFA was described for *S. oneidensis* and *S. putrefaciens* (based on the presence of the FadR regulon and the Des enzyme), however, the PUFA synthesis was shown only for *S. oneidensis* and *S. baltica* [91,92]. Our transcriptome analysis revealed that the genes involved in biosynthesis of UFA and PUFA are under cold stress mediated regulation. We found that expression of a *fabA* ortholog (Shew185_2605) had increased after 180 min of the cold stress onset, which correlated with induced expression of *fadR* (Shew185_1771) and decreased expression of *fabR* (Shew185_2605) (Figure 6, Appendix A). These observations indicate that the process of UFA synthesis is induced and upregulated as a response to cold stress. Analogous results were recently shown for *S. putrefaciens* [92]. Interestingly, expression of the *des* gene (Shew185_2748) was decreased (Appendix A) which again correlates with the *S. putrefaciens* data, suggesting that the anaerobic pathway of UFA synthesis plays a major role in the cold-induced modification of membrane composition for the *Shewanella* strains. PUFA production mediated by the *pks* and *pfaB* genes (Shew185_1419 and Shew185_1418, respectively) is also induced, as the expression of these genes was already elevated after 90 min of cold stress (Appendix A). Thus, our data present evidence that *S. baltica* adjusts to the low temperature by maintaining its membrane fluidity by altered membrane composition.

### 2.9. Changes in Amino Acid Metabolism and Translation-Related Pathways in Response to Cold Stress

After 90 min of cold stress, we observed a significant downregulation of metabolic pathways of several amino acids in *S. baltica*, including arginine, valine, leucine and isoleucine, phenylalanine, tyrosine, tryptophan, arginine, proline, histidine, phenylalanine, and beta-alanine metabolism. Downregulation of these pathways was still present at 180 min; moreover, the downregulated pathways also included glycine, serine, threonine, lysine and tyrosine metabolism (Appendix A). Most of these pathways were characterized with downregulation of 50% or more genes assigned to each pathway; valine, leucine and isoleucine metabolism appeared to be the most affected pathway with over 80% of assigned genes being significantly downregulated (Appendix A). Interestingly, initially mostly non-polar amino acid metabolism was transcriptionally downregulated, but later polar amino acid metabolism was affected as well. It was previously shown that *S. oneidensis* downregulates genes of amino acid biosynthesis under cold stress conditions [23]. A study of cold stress-induced proteomic response confirmed that a variety of *S. oneidensis* proteins used in amino acid synthesis was also downregulated [93]. Results of functional annotation of the differentially expressed genes from *S. piezotolerans* showed upregulation of amino acid transport and metabolism-related genes upon heat stress, and downregulation upon cold stress [94]. Contrary, analysis of cold stress response in *T. tengcongensis* revealed that among differentially expressed genes, those involved in synthesis of branched amino acids were upregulated [55]. Interestingly, along with downregulation of amino acid metabolism, we observed upregulation of some translation-related pathways. Our results of functional annotation of differentially expressed genes from *S. baltica* showed upregulation of the aminoacyl-tRNA biosynthesis pathway (Appendix A); also, expression of a set of aminoacyl synthetases was affected after the onset of cold stress (Appendix A). Gene expression of a set of tRNAs for Ala, Ile, Leu, Ser, Glu and Met was also upregulated (Appendix A). In addition, the category ‘ribosome’ was upregulated as well (Appendix A). Transcriptional level of genes required for the biosynthesis of many amino acids was previously found to be reduced during cold stress response in *B. subtilis* [95]. Induction of several genes encoding ribosomal proteins was also found in this bacterium [95]. However, in contrast to our results, genes encoding some of the tRNA synthetases, *aspS*, *hisS*, *metS* and *thrS*, were repressed in *B. subtilis* [95].

Intriguingly, it was reported that psychrophilic strains of *Shewanella* spp. and other cold-adapted γ-proteobacteria have a decreased alanine, proline, glycine and arginine content in their proteomes in comparison to the warm-adapted *Shewanella*, whereas the content of amino acids, such as lysine, isoleucine and asparagine was found to be increased in the cold-adapted proteins [96]. This characteristic amino acid profile was suggested to be beneficial for protein structural flexibility at low temperatures. In our study, we observed transcriptional decrease in expression of several amino acids metabolism-related genes, including those involved in the alanine, proline, glycine, and arginine metabolic pathways. Although lysine or isoleucine biosynthetic pathways in our study were found to be downregulated, we observed that the degradation pathway of these two amino acids was downregulated as well. It seems that in *S. baltica*, the metabolism of certain amino acids can be modulated in order to obtain a well-fitting composition of proteins upon the temperature drop. Thus, we hypothesize that bacterial adaptation to cold may be considered not only as an evolutionarily-gained feature of cold-adapted bacteria, as it was previously reported, but also as an efficient switching mechanism ensuring fast and accurate adaptation to temporal cold conditions. Nevertheless, this hypothesis needs to be verified.

## 3. Materials and Methods

### 3.1. Bacteria and Growth Conditions

*S. baltica* M1 strain employed in this study was isolated from surface water of the Gulf of Gdańsk and its genome was sequenced previously [97]. Bacteria were cultured in chemostats in the Instant Ocean™ Synthetic Sea Salts (Fisher Scientific, Ottawa, ON, Canada) medium in a total volume of 150 mL, at 20 °C, with the growth rate of µ = 0.2. The cold stress was achieved by shifting temperature of the media down to 8 °C. At time 0, and after 90 and 180 min of cold stress, samples (5 mL) were withdrawn and equal volumes of RNAprotect Bacteria Reagent (Qiagen, Hilden, Germany) were added. After 5 min of incubation at room temperature, samples were centrifuged (10 min; 5000× *g*), the pellet was resuspended in 200 µL of standard Tris-EDTA buffer and frozen in liquid nitrogen.

### 3.2. RNA Isolation and Sequencing

RNA was isolated from chemostat-culture samples using the RNeasy Mini Kit (Qiagen, Hilden, Germany) according to the manufacturer’s recommendation. The quality and concentration of the isolated RNA was assessed by using Bioanalyzer (Agilent Technologies, Santa Clara, CA, USA). The library construction and RNA sequencing was performed through the RNA-Seq Service (Macrogen Europe, Amsterdam, Netherlands). Briefly, a paired-end, strand-specific library was constructed from RNA depleted of ribosomal RNA (Ribo-Zero rRNA Removal Kit (Bacteria) (Illumina, San Diego, CA, USA)) using TruSeq Stranded Total RNA Sample Prep Kit (Illumina, San Diego, CA, USA). The data discussed in this publication have been deposited in NCBI’s Gene Expression Omnibus [98] and are accessible through the GEO Series accession number GSE150927 (https://www.ncbi.nlm.nih.gov/geo/query/acc.cgi?acc=GSE150927).

### 3.3. Bioinformatics Analysis

The raw sequence data were quality filtered using the fastp software [99]. Filtered fastq files were used for mapping to the *S. baltica* OS185 chromosome (accession number: NC_009665.1) using Rockhopper version 2.0 software with default parameters [100]. Despite ribodepletion, between 12–23% reads were mapped to ribosomal RNA genes. This resulted in 783× coverage for protein coding genes, on average. Summary statistics are available in Appendix A. Rockhopper was used for subsequent normalization and differential gene expression analysis. The results table generated by the software was further processed and filtered using R. KEGG enrichment analysis was performed using the KEGGprofile package [33].

### 3.4. RT-qPCR Assay and Data Analysis

Synthesis of cDNA was performed in a total volume of 20 µL; the reaction contained 1 µg of total RNA, 4 µM random hexamers (Invitrogen by Thermo Fisher Scientific, Carlsbad, CA, USA), 1 mM dNTP Mix (Invitrogen by Thermo Fisher Scientific, Carlsbad, CA, USA), 1X RT buffer, 5 mM DTT, 1.25 U of RNase Inhibitor and 0.5 U of AMV Reverse Transcriptase Native (EurX, Gdańsk, Poland), according to the manufacturer’s instructions. Priming of RNA-hexamers-dNTP was performed at 65 °C for 5 min and then was chilled on ice. Reverse transcription was performed in a thermocycler under the following conditions: 20 °C for 10 min (random primer extension), 50 °C for 50 min (cDNA synthesis) and 75 °C for 10 min (inactivation). Reverse transcriptions were performed in duplicate. Samples of cDNA were diluted 1:20 in water, and 4 µL aliquots were used as a template in subsequent qPCR reactions.

Primers for qPCR of target and reference genes were designed using the Clone Manager 10 software (Sci Ed Software LLC, Westminster, CO, USA) and synthesized by Sigma-Aldrich (St. Louis, MO, USA) custom service; primer sequences are available in Appendix A. All PCR products obtained with the designed primers were checked for consistency with the expected sequence by Sanger sequencing (Macrogen Europe, Amsterdam, Netherlands). Real-time qPCR amplification was performed in a total volume of 10 µL using the SG qPCR Master Mix (2X) kit (EurX, Gdańsk, Poland) according to the manufacturer’s instruction, with the 0.5 µM final concentration of each primer. Reactions were run in the LightCycler^®^ 480 Instrument II (Roche Diagnostics Ltd., Rotkreuz, Switzerland) at the following conditions: 95 °C for 2 min (initial denaturation), followed by 35–45 cycles of 94 °C for 15 s, 53 °C for 30 s, and 72 °C for 30 s. Melting curve analysis was performed between 65 °C and 95 °C. Crossing threshold (Ct) values were generated by the LightCycler^®^ 480 Software version 1.5 (Roche Diagnostics GmbH, Rotkreuz, Switzerland). We assumed that the optimal Ct values should be in the range of 20–32 cycles; if lower or higher Ct values were obtained, volumes of cDNA template were adjusted and qPCR amplifications were repeated. Each cDNA template was amplified twice with two technical replicates for each assay. Negative controls, such as reactions without reverse transcriptase or without cDNA template, were included in our analysis. For validation of reference genes, we used a set of 10-fold dilutions (ranging from 1 to 1:10,000) of cDNA as a template for qPCR. The stability and validity of five candidate reference genes for normalization of RT-qPCR results were determined by the BestKeeper version 1 tool [101] using raw Ct values. The 2^−∆∆Ct^ algorithm was used to calculate the relative gene expression [102]. All results were checked for outliers with Iglewicz and Hoaglin’s two sided test for multiple outliers [103]; outlier criterion was set at the modified Z-score ≥ 3.5.

## 4. Conclusions

Here, we present evidence that *S. baltica* undergoes global transcriptional changes in order to adapt to cold stress. The most pronounced response was massive downregulation of approximately 70% of differentially expressed genes, indicating a general shutting down of many energy-consuming pathways to increase survival. However, certain pathways were upregulated, and their roles generally correspond with stress adaptation and in some extent coincide with analogous responses observed in other *Shewanella* genus representatives. In particular, cold induced proteins’ gene expression was increased, and the fatty acids synthesis pathways were adjusted to alter membrane composition to higher content of unsaturated fatty acids to preserve membrane fluidity. Interestingly, global transcription reprogramming was observed in cold-stressed *S. baltica*: expression of the *rpoE* gene, encoding the sigma E factor, was induced in response to the envelope and oxidative stress, while the housekeeping sigma factor, *rpoD,* expression decreased. We also observed an increase in the flagellar genes’ expression indicating that *S. baltica* intensifies its motility in response to cold stress. Modulation of amino acid metabolism and protein translation network is another evident response of *S. baltica* to the temperature drop. Alterations in amino acid profiles during cold stress are apparently common in bacterial world, thus it may be assumed as a universal response mechanism ensuring adequate protein structural flexibility, stability and activity at low temperatures. Transcriptional downregulation of type II secretion system observed in our study will definitely affect the export of proteins outside the cell via this route. This phenomenon of complete transcriptional switch off of one specific secretion system needs further investigation to elucidate its role in bacterial survival and adaptation capacity at low temperatures. Our results also indicate that cold stress may stimulate *S. baltica* to prepare for potentially upcoming anaerobic conditions, which are common at low temperatures in the complex marine environment. We detected a significant upregulation of expression of genes assigned to the assimilatory branch of the sulfate reduction pathway, but we did not observe a transcriptional switch to anaerobic respiration at the transcriptional level, perhaps due to a relatively short period of cold stress in our experimental design. However, as downregulation of genes encoding degradation of thiosulfate and tetrathionate was observed, we hypothesize that these two compounds might eventually serve as electron acceptors in alternative respiration pathways, which were previously described for the *Shewanella* genus.

*Shewanella* species are considered as the main players in denitrification and bioremediation processes; they are also known for their spoilage and decomposition potential. Thus, revealing the mechanisms of *S. baltica* response to low temperatures is important not only for the general knowledge of bacterial cold stress adaptability, but it is also of great importance for the marine environmental studies, as well as for biotechnology, with focus on e.g., bioremediation, and even for the fishery industry.

## Figures and Tables

**Figure 1 ijms-21-04338-f001:**
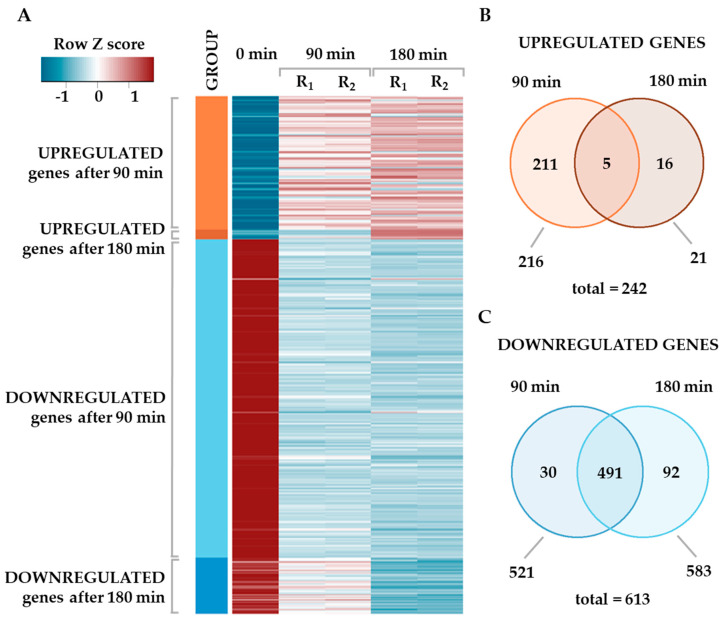
Changes in the global gene expression upon cold stress. (**A**) Heatmap presenting changes in gene expression for significantly altered genes (|log_2_FC| > 1 and adjusted *p* value <0.01). Column 0 stands for the control sample; replicates for 90 and 180 min time points are labeled R_1_ and R_2_. Gene expression data are presented as Z-score transformed, scaled in rows values. (**B**,**C**) Venn diagrams representing overlap between differentially expressed genes at both time points.

**Figure 2 ijms-21-04338-f002:**
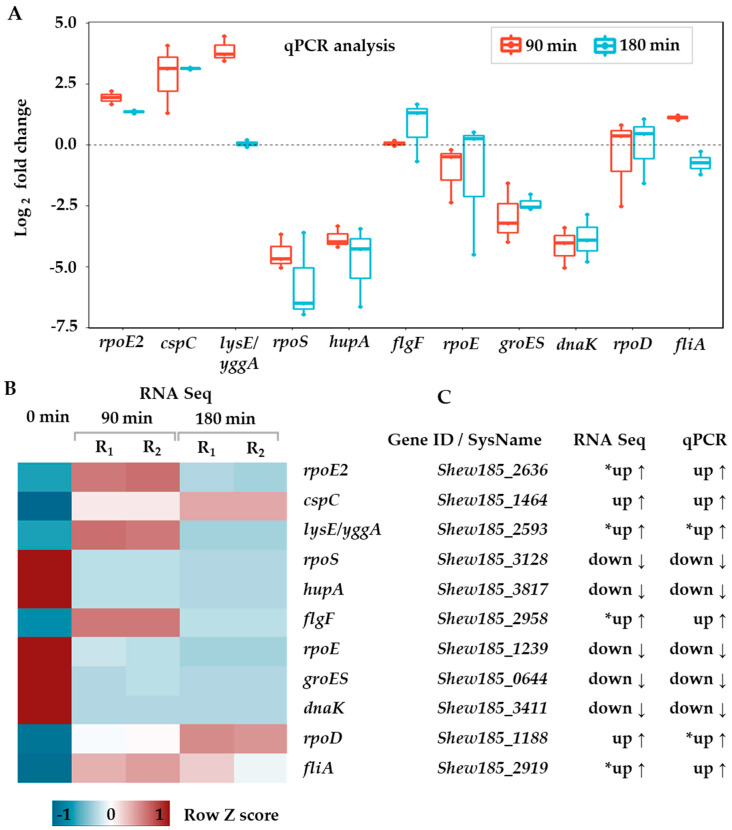
Validation of RNA sequencing results with RT-qPCR. (**A**) Log_2_ fold change of relative gene expression measured by RT-qPCR. Data were normalized to three reference genes (for details see the Materials and methods, Section 3.4). Each point represents average of four technical repeats for a biological replicate. Red and blue colors mark 90 and 180 min time points, respectively. (**B**) Heatmap showing row Z-score transformed, row scaled expression for corresponding genes. Column 0 stands for control sample for 0 min time point; replicates for 90 and 180 min time points are labeled R_1_ and R_2_. (**C**) Direction of changes in respect to control (time 0) for corresponding RNA-seq and RT-qPCR assays. Asterisk (*) marks the expression change categorized as up although in fact it is upregulated only after 90 min while downregulated after 180 min.

**Figure 3 ijms-21-04338-f003:**
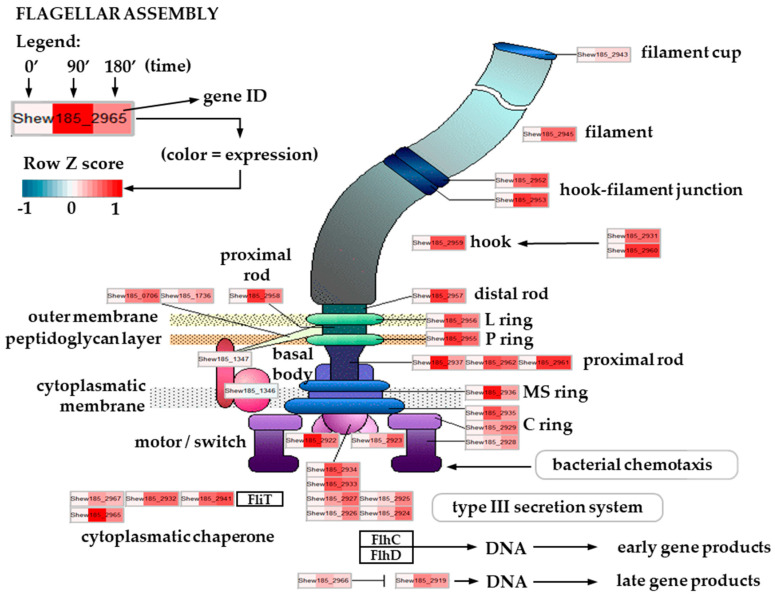
Expression changes in *S. baltica* flagellar assembly pathway upon cold stress. Scheme of flagellar assembly pathway (source: Kyoto Encyclopedia of Genes and Genomes (KEGG) database). Tripartite heatmap boxes show row Z-score scaled normalized expression values for corresponding genes upon cold stress at 0, 90 and 180 min time points, respectively.

**Figure 4 ijms-21-04338-f004:**
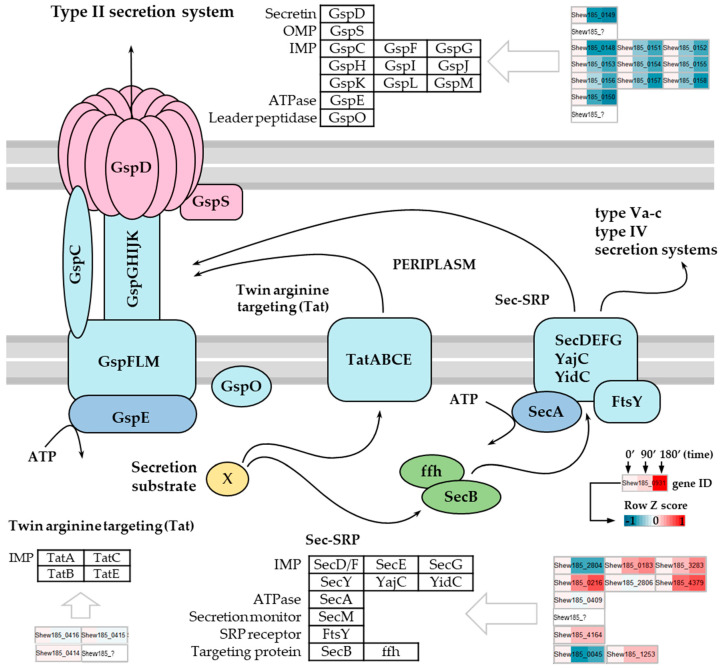
Expression changes affecting the *S. baltica* type II secretion system upon cold stress. Scheme of type II secretion system with twin-arginine targeting (Tat) and SecA and signal recognition particle (Sec-SRP) translocons (source: KEGG database). Tripartite heatmap boxes show row Z-score scaled normalized expression values for corresponding genes upon cold stress at 0, 90 and 180 min time points, respectively. Question mark denotes that the gene is not annotated in *S. baltica* OS185.

**Figure 5 ijms-21-04338-f005:**
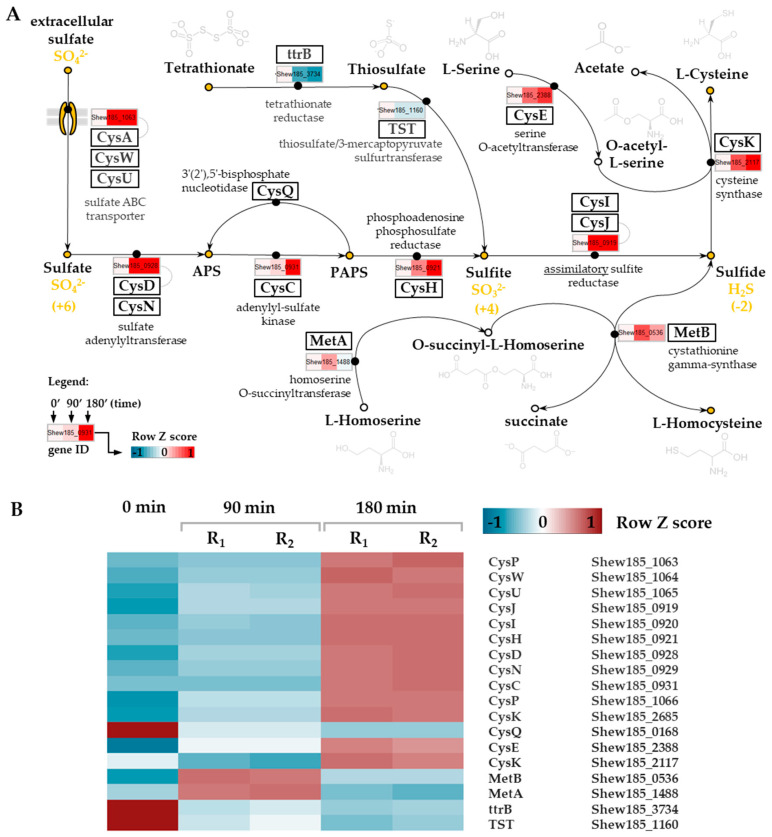
Expression changes in *S. baltica* sulfur metabolism processes upon cold stress. (**A**) Simplified scheme of sulfur assimilatory pathway and a fragment of serine, homoserine and cysteine metabolism pathways are presented (based on: KEGG database). Tripartite heatmap boxes, fixed on affected pathways (full black circles), show row Z-score scaled normalized expression values for corresponding genes upon cold stress at 0, 90 and 180 min time points, respectively. Names of specific enzymes or subunits are shown next to the relevant gene names (black boxes). Yellow circles represent sulfur-containing compounds. Oxidation states and formulas (in yellow) are shown for key inorganic sulfur compounds—sulfate, sulfite and sulfide. (**B**) Heatmap showing row Z-score scaled normalized expression for corresponding genes. Column 0 stands for control sample; replicates for 90 and 180 min time points are labeled R_1_ and R_2_.

**Figure 6 ijms-21-04338-f006:**
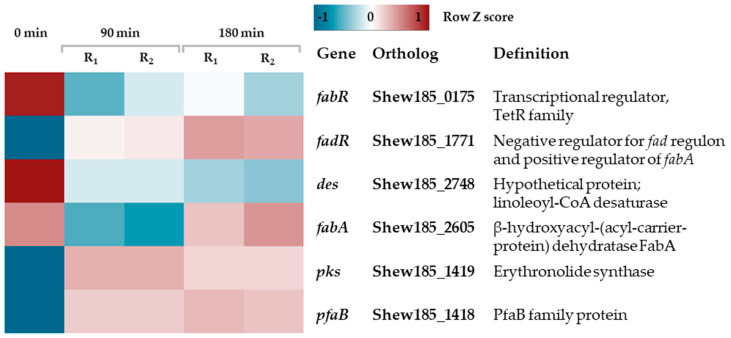
Expression changes in *S. baltica* fatty acid metabolism upon cold stress. Heatmap showing row Z-score scaled normalized expression for corresponding genes. Column 0 stands for control sample; replicates for 90 and 180 min time points are labeled R_1_ and R_2_.

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
