# Peer review of "Adaptation of the Marine Bacterium *Shewanella baltica* to Low Temperature Stress"

_ijms, 2020, doi:10.3390/ijms21124338_

Round 1

Reviewer 1 Report

The article studies gene expression at low temperatures. The results are interesting and provide new data on the response of Shewanella baltica to cold stress. The paper is pubishable.

Author Response

Response to Reviewer 1 Comments

Esteemed Reviewer,

We are grateful for the evaluation of our manuscript and your opinion that it is publishable with no revisions suggested.

Reviewer 2 Report

The manuscript by Kloska et al., is a description of RNA-seq results of the bacterium Shewanella baltica in response to cold stress. The authors were able to identify several interesting areas of response to cold and have hypothesised how these may be related to the issue of food spoilage. Overall, the experiments and analysis are well performed, with the main issue being the required to fix many of the language issues – otherwise the manuscript is great. I only have some other minor comments below.

Line 92: While it is good that this sentence is referenced, I think the authors can still list the rules of good practice that they followed so that it is very clear to the reader.

Figure 1: I am confused by Figure 1A. Should the 0min column be white, since all these genes are the 0 minute and therefore is the reference for comparison – ie. 0 fold change. What are the fold change in relation to for the 0 min column?

Figure 2: Again, I am not sure why the 0 min column has a fold change. Further, While I think it is very important that the authors confirmed their RNA-seq findings using RT-qPCR, this results did not actually result in any new analysis or findings, and therefore I think the figure does not need to be in the main text and can simply be moved to the supporting information.

Line 194: ECF should be written in long form for the first time on line 192.

Line 230-231. Needs a reference to the line ‘are widespread in nearly all tested bacteria’.

Figure 3. Perhaps this should be separated into two separated figures. I am unsure how figure 3A is directly related to 3B. Also for figure 3A, what unit is on the x axis?

Author Response

Response to Reviewer 2 Comments

Esteemed Reviewer,

We are grateful for the evaluation of our manuscript, as well as for suggestions and recommendations, which lead to the improvement of the quality of our article. Please find bellow our point-by-point answers to your comments. 

Point 1: Overall, the experiments and analysis are well performed, with the main issue being the required to fix many of the language issues – otherwise the manuscript is great.

Response 1: The revised version of our manuscript was checked by an English speaking colleague. All language corrections are highlighted in the manuscript using “Track changes” function of the Word processor.

Point 2: Line 92: While it is good that this sentence is referenced, I think the authors can still list the rules of good practice that they followed so that it is very clear to the reader.

Response 2: Actually we shortly mentioned these rules by stating how our data were filtered in the sentence describing filtering strategy. However, for emphasis of the good practice roles, we propose following modification:

Lines 95–97 (highlighted in yellow): The main role was to filter genes with distinct changes, therefore, our main data set was composed of genes which had at least a two-fold change in expression (|log2FC| > 1) and adjusted p value < 0.01 (Supplementary Table S1).

Point 3: Figure 1: I am confused by Figure 1A. Should the 0min column be white, since all these genes are the 0 minute and therefore is the reference for comparison – i.e. 0 fold change. What are the fold change in relation to for the 0 min column?

Response 3: Heatmap in Figure 1 is presented as Z-score which is not the same as fold change. Z-score is a result of scaling procedure which is commonly implemented in heatmap drawing software or packages (here, heatmap.2 function in gplots package for R). Z-score instead of raw values is commonly used for simple visualization of presented data. In a heatmap with a wide spread of values, the color intensity will be the same for a range of values thus, it will not show any differences in colors despite significant fold changes. In fact, for this data set, the heatmap will be white with some blue and red colors only for the genes that changed the most. On the contrary, in row Z-score scaling, the mean expression value from all tested conditions is subtracted from values obtained for every condition (in a row). Thus, even in time assigned to 0 min will have a Z-score value different from 0 an appropriated color will be assigned. Conditions with and average expression will have Z-score values close to 0, conditions with the expression lower than the average will have negative Z-score values and conditions with expression higher than the average will have positive Z-score values. As we stated Z-score transformation is a mathematical scaling procedure used for the majority of heatmaps and in our manuscript, we clearly state that on every heatmap we present the row Z-score and not the actual fold change or log2 transformed fold change.

For clarity, we have made the following modification in Figure 1 caption (lines 115-116, highlighted in yellow): Gene expression data are presented as Z-score transformed, scaled in rows values.

Point 4: Figure 2: Again, I am not sure why the 0 min column has a fold change. Further, While I think it is very important that the authors confirmed their RNA-seq findings using RT-qPCR, this results did not actually result in any new analysis or findings, and therefore I think the figure does not need to be in the main text and can simply be moved to the supporting information.

Response 4: Figure 2 caption also has been modified as discussed above: Heatmap showing row Z-score transformed, row scaled expression for corresponding genes. (line 135, highlighted in yellow).

Further, we understand Reviewer's concerns about presenting RT-qPCR data here. However, we feel like this is a very important experiment regarding our RNA-seq data. First, there are many studies where RT-qPCR validation is often performed only for genes with the greatest expression changes obtained in RNA-seq experiment; also, many studies show only a modest concordance of RT-qPCR data with RNA-seq data. Secondly, here we discuss the theoretical power of our analysis which, accordingly to predictions, is prone to high false discovery rate, yet, our RT-qPCR analysis showed that expression changes for randomly selected amplicons are concordant with RNA-seq predictions. Therefore, we would like to retain Figure 2 in its current form since the discussion of statistical power and false discovery rate for our experiments is crucial for further conclusions.

Point 5: Line 194: ECF should be written in long form for the first time on line 192.

Response 5: The long form for ECF (extracytoplasmic function factors) is included in a line were the abbreviation appears for the first time (now: line 198, highlighted in yellow).

Point 6: Line 230-231. Needs a reference to the line ‘are widespread in nearly all tested bacteria’.

Response 6: A relevant reference has been added to the sentence (now: line 237, highlighted in yellow) – [46] Phadtare, S.; Alsina, J.; Inouye, M. Cold-shock response and cold-shock proteins. Curr. Opin. Microbiol. 1999, 2, 175–180.

Point 7: Figure 3. Perhaps this should be separated into two separated figures. I am unsure how figure 3A is directly related to 3B. Also for figure 3A, what unit is on the x axis?

Response 7: Figure 3A has been moved to supplementary materials as we agree with the Reviewer, that indeed it is not directly related to the scheme of flagellar assembly (originally Figure 3B and now assigned as Figure 3, line 290). The removed graph (now: Supplementary Figure S1), and similar graphs for other time points, show expression changes for all genes assigned to particular KEGG pathways in a form of a box-plot (each point is a gene from that pathway). The values on the x-axis are “Normalized counts” (originally we assigned it as “Normalized expression”). We have corrected the description of the x-axis on the former Figure 3A (as well as on all the graphs in Supplementary Figure S1) and made necessary adjustments in the corresponding Figure 3 and Figure S1 captions.